# Emphasizing Semantic Consistency of Salient Posture for Speech-Driven Gesture Generation

## ABSTRACT

Speech-driven gesture generation aims at synthesizing a gesture sequence synchronized with the input speech signal. Previous methods leverage neural networks to directly map a compact audio representation to the gesture sequence, ignoring the semantic association of different modalities and failing to deal with salient gestures. In this paper, we propose a novel speech-driven gesture generation method by emphasizing the semantic consistency of salient posture. Specifically, we first learn a joint manifold space for the individual representation of audio and body pose to exploit the inherent semantic association between the two modalities, and propose to enforce semantic consistency via a consistency loss. Furthermore, we emphasize the semantic consistency of salient postures by introducing a weakly-supervised detector to identify salient postures, and reweighting the consistency loss to focus more on learning the correspondence between salient postures and the high-level semantics of speech content. In addition, we propose to extract audio features dedicated to facial expression and body gesture separately, and design separate branches for face and body gesture synthesis. Extensive experiments and visualization results demonstrate the superiority of our method over the state-of-the-art approaches.

## CCS CONCEPTS

• **Computing methodologies → Animation**.

## KEYWORDS

speech-driven gesture generation, semantic consistency, neural generative model, multi-modality

## 1 INTRODUCTION

The task of speech-driven gesture generation aims to synthesize a sequence of gestures line with the given speech signal, which has a wide range of application scenarios, including online service [20, 22], virtual avatar animation [21, 39] and human-machine interaction [6, 30, 31]. Compared with lip motion generation, speech-driven gestures are more implicit and metaphoric, which makes the gesture generation task a non-trivial challenge.

Considering the significant modality gap between speech and gestures, traditional works [8, 11, 19, 33] tackle the problem through

*ACM MM, 2024, Melbourne, Australia*
© 2024 Copyright held by the owner/author(s). Publication rights licensed to ACM.
ACM ISBN 978-x-xxxx-xxxx-x/YY/MM
https://doi.org/10.1145/nnnnnnn.nnnnnnn

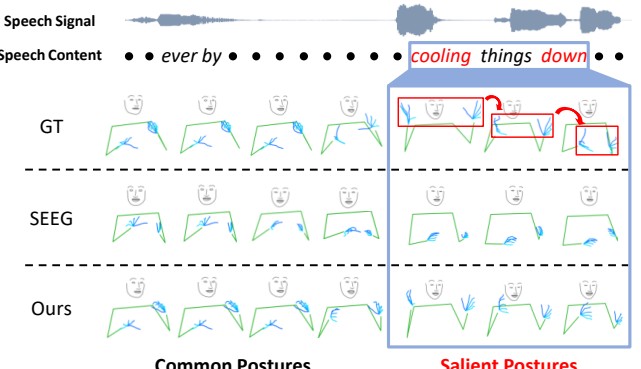

**Figure 1: Salient postures indicate the large pose movements associated with the high-level semantics of speech content, e.g., *cooling down*, which are hard to be generated. Our method can synthesize more realistic gestures than SEEG [21] by emphasizing semantic consistency of salient postures.**

rule-based generation approaches, which establish the deterministic correspondence between audio syllables and gesture sets. Such methods ignore the intrinsic connection between different modalities and suffer from poor naturalness. Recent data-driven methods [21, 24] achieve better performance by utilizing deep neural networks to extract audio representation of different semantic granularities, which is then decoded to generate a holistic gesture sequence. However, these methods suffer from the following major weaknesses: (1) their pipelines are straightforward and cannot effectively achieve semantic consistency between the speech content and the synthesized gestures, especially for the postures with large movement scope. (2) They treat facial expressions and body movements as a whole and simultaneously synthesize them with a single pipeline, which typically leads to poor synchronization between facial expressions and speech.

Therefore, our primary goal is to enhance the correspondence between the generated gestures and the semantics in the speech content. From a human gesture study [12], we identify one significant observation: postures with large movement scope in the sequence correspond to strong and rich semantic information of speech audio, and postures with slight movement scope correspond to weak semantics of speech audio. As illustrated in Figure 1, the gesture sequence of opening and raising the arms, then folding and lowering them is closely related to the phase *cooling down*. For clarity, we define salient postures as the postures with large movement scope, i.e., the movements in which the shoulders and arms have a relatively large amplitude of motion, which are often related to strong semantics of the speech content. Salient postures in gesture sequences tend to be significant in conveying the intention and emotion of speakers, and therefore the exact correspondence relationship between salient postures and speech with strong semantics will contribute to the vivid and realistic gesture generation.

Inspired by the above discussions, we propose a novel speech-driven gesture generation method by emphasizing semantic consistency of salient posture. To fully exploit the inherent semantic association between audio and gesture, we first learn a joint manifold space for the representations of audio and body pose to establish a mapping between the two modalities. Through the consistency loss, we ensure that the audio and pose features are close to each other in the shared joint-embedding space and represent similar semantic information.

Furthermore, we propose to **emphasize the semantic consistency of salient postures**, i.e., postures with large movement scope that often correspond to strong semantics. We design a weakly-supervised salient posture detector to identify salient postures, which utilizes a temporal relation module to mine long-range temporal dependencies among the pose features and predicts frame-level saliency score. We then use the saliency score to reweight the consistency loss to enforce a stronger alignment between the salient postures and corresponding audios in the joint embedding.

In addition, facial expressions, especially lip motions, mainly rely on articulation-related acoustic features; while body gestures are closely correlated with strong semantics in the speech content. Therefore, we extract audio features dedicated to facial expressions and body gestures separately, and synthesize facial expressions and body gestures with separate branches. Meanwhile, we enforce the temporal alignment between the audio features extracted in the face branch and the body branch, which effectively improves synchronization and naturalness between the face and body parts. Our main contributions can be summarized as follows:

- We propose a novel speech-driven gesture generation framework with an emphasis on semantic consistency of salient posture. We introduce the joint manifold space to learn the inherent semantic association between audio and gesture modalities and enforce semantic consistency via a consistency loss.
- We emphasize the semantic consistency of salient postures by introducing a weakly-supervised detector to identify salient postures, and reweighting the consistency loss based on saliency score to enforce a stronger alignment in the joint manifold space.
- Observing that facial expressions rely on articulation-related audio features while body gestures rely on semantic-related audio features, we propose to extract separate audio features for face and body, and design separate branches for face and body gesture synthesis.

## 2 RELATED WORK

**Audio-Visual Cross-Modal Learning.** Multi-modal machine learning aims to train models capable of processing and relating information from multiple modalities. Recent works [1, 2, 35] encode all the modalities into a common representation space. Language2Pose [2] learns a joint embedding of text and pose. Ahuja *et al.* [1] maps the learned style embedding along with audio into a joint gesture space. Trimodal [35] utilizes separate representations for different modalities and handles the alignment between speech and gesture explicitly. Compared with these methods, our method focuses on identifying the strong semantic correlations between speech audio

and visual gestures to synthesize more natural-looking and vivid gesture sequences.

**Speech-Driven Gesture Generation.** Synthesizing consistent and natural gestures in line with speech is becoming popular in the field of multi-modal generation. With the development of deep learning, recent works [3, 13, 29, 32] leverage deep neural networks to generate more natural gesture sequences. Audio2Body [32] learns the correlation between audio features and body landmarks through an LSTM network and concentrates on predicting body motion from specific music like piano and violin. S2G [13] incorporates generative adversarial learning into the regression-based prediction model to enhance the naturalness of generated results. MoGlow [3] adapts the probabilistic model to this task to learn the distribution of gesture motions and can effectively take control over gesture styles. SDT [29] learns template vectors to provide extra information for gesture generation and transform the one-to-many ambiguous regression problem into a deterministic conditional regression problem. However, these methods ignore the semantic association of different modalities and fail to deal with habitual and salient gestures. Our method utilizes joint manifold space to model the mapping function and design a salient posture detector to maintain semantic consistency of salient gestures.

**Anomaly Detection.** Video anomaly detection aims to identify abnormal events in the video that do not match the normal behaviors. Early works [16, 18, 27, 28] focus on detecting anomalies through manually extracting features and modeling the anomalies. Recently, several deep learning-based methods [14, 23, 25, 26, 37] have achieved significant performance improvement. Reconstruction-based methods [14, 26, 37] utilize the autoencoder trained on normal datasets and detect anomalous frames which are difficult to be reconstructed well. Prediction-based methods [23, 25] predict future frames and utilize prediction error as an indicator to determine anomalies, considering the frames with large prediction errors as anomalies. Different from these methods, we design a weakly-supervised salient posture detector to identify anomaly gestures only under the weak supervision of video-level labels. and reweight the cross-modal association based on the predicted saliency score.

## 3 METHOD

### 3.1 Overview and Notations

To fully empower the learning of semantic association between speech and gesture, we propose a novel speech-driven gesture synthesis method that emphasizes semantic consistency of salient postures, i.e., postures with large movement scope that often correspond to strong semantics. Our overall architecture is shown in Figure 2. Our model first learns a joint manifold space for different representations of audio and body pose to explore a finer mapping between two modalities. Then, we emphasize the semantic consistency of salient postures by introducing a weakly-supervised detector to identify salient postures, and enforcing a stronger alignment for the salient postures in the joint manifold space. In addition, observing that facial expressions rely on articulation-related audio features while body gestures rely on semantic-related audio features, we extract separate audio features for face and body, and design separate branches for face and body gesture synthesis.

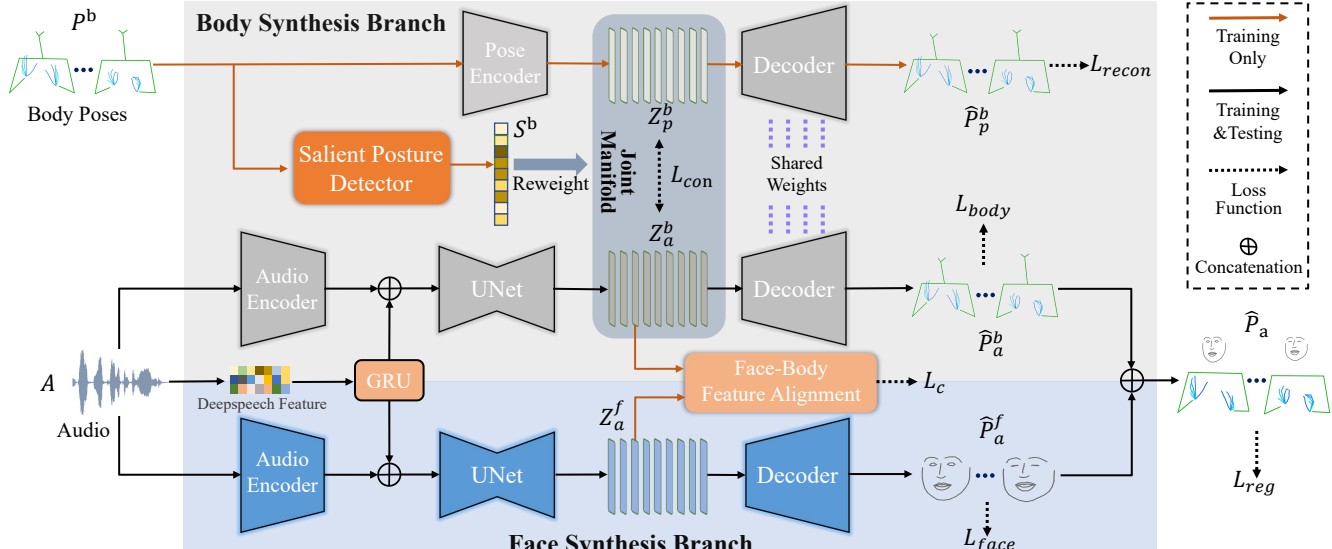

Figure 2: The overall architecture of our proposed method, which consists of two branches including body synthesis branch and face synthesis branch. In the body synthesis branch, our model learns a joint manifold space of representations to enforce semantic consistency by employing a dual-path structure, which contains the upper reconstruction path and the lower speech-driven generation path. Furthermore, the salient posture detector is designed to identify salient gestures and reweight the consistency loss. We then generate synchronized facial expressions using face synthesis branch. Finally, we fuse the generated results of two branches to obtain the entire gesture sequence.

Given an audio clip $A = [A_1, \ldots, A_T]$, speech-driven gesture synthesis aims to generate a pose sequence $P = [P_1, \ldots, P_T]$ synchronized with the audio, where the length of sequence is $T$ and $P_t$ is the pose at the $t^{\text{th}}$ frame. Each $P_t$ is defined as the coordinates set of $J$ upper-body keypoints in a frame, including face, hands, and arms. The major challenge of synthesizing realistic gestures is how to establish the reasonable mapping: $A_{1:T} \rightarrow P_{1:T}$. Due to the discrepancy of mapping property of different parts, we decompose the entire pose $P_t$ into body part $P_t^b$ and face part $P_t^f$. The whole gesture sequence $P$ can be formulated as: $P = P^b \oplus P^f$.

For the generation of body poses, we first utilize two feature extractors to extract audio and pose latent codes respectively. Then, we use a pose decoder to separately generate poses from these two latent codes. The output of our pose decoder is a sequence of $T$ frames. We denote the pose sequence generated from pose latent codes as $\hat{P}_p^b$, and the pose sequence generated from audio latent codes as $\hat{P}_a^b$. Then, for facial expressions synthesis, we extract different audio features (articulation-related) and directly generate a sequence of $T$ frames $\hat{P}_a^f$. Eventually, by fusing the synthesis results of the two branches, we obtain the final output of the entire gesture sequence $\hat{P} = \hat{P}_a = \hat{P}_a^b \oplus \hat{P}_a^f$.

## 3.2 Joint Manifold Space for Speech and Gesture

Due to the high randomness of body motion, our model learns a multi-modal joint manifold space between audio and body pose to explore the semantic correlation between audio and pose representations. To form the shared joint-embedding space, as shown in the upper part of Figure 2, we employ a dual-path architecture that consists of two parallel pipelines: the reconstruction path for

body pose and the speech-driven gesture generation path. The reconstruction path takes as input the real pose sequence of body part $P^b$, and uses a pose encoder $Enc_p$ consisting of two GRUs to obtain a pose feature in the joint manifold space,

$$Z_p^b = Enc_p(P^b), \quad (1)$$

where $Z_p^b \in \mathbb{R}^{T \times D}$ is the latent code of body pose and $D$ represents the dimension of the latent code.

For the speech-driven gesture generation path, we first utilize the audio encoder $Enc_a$ to encode the mel-spectrogram of audio and concatenate it with the feature code $f_d$ extracted by the *DeepSpeech* [4] model, which is pre-trained by large numbers of audio-transcript pairs. The integration of high-dimensional representations of *DeepSpeech* can provide richer semantic information for follow-up gesture generation. We further map the concatenated feature to an audio feature in the joint manifold space using 1D UNet translation network,

$$Z_a^b = UNet(Enc_a(A) \oplus f_d), \quad (2)$$

where $Z_a^b \in \mathbb{R}^{T \times D}$ is the latent code of audio corresponding to body part.

To guarantee that $Z_p^b$ and $Z_a^b$ lie close to each other in the shared joint-embedding space and represent similar semantic information, we propose the consistency loss to constrain the latent codes. Similar to the cosine similarity function, the consistency loss calculates the cosine similarity between the body pose latent code and audio latent code:

$$L_{con} = \sum_{t=1}^{T} (1 - \frac{Z_{p,t}^b \cdot Z_{a,t}^b}{\max(||Z_{p,t}^b|| \cdot ||Z_{a,t}^b||, \epsilon)}), \quad (3)$$

Figure 3: The detailed structure of proposed salient posture detector. We take as input the real body poses $P^b$ and extract the initial feature $X$ using the ConvNet. Then, $X$ is fed into the temporal relation module to obtain the interaction feature $Y$. We utilize a classifier to map $Y$ to the 1D salient score $S^b$, which is used to reweight the consistency loss of joint manifold training.

where $Z_{p,t}^b$ and $Z_{a,t}^b$ are the $t^{\text{th}}$ value of latent codes along time axis respectively, and $\epsilon$ is a fairly small positive scalar.

Then, the decoders of the reconstruction path and generation path learn to generate the pose sequences $\hat{P}_p^b$ and $\hat{P}_a^b$ from the above two latent codes,

$$\hat{P}_p^b = Dec(Z_p^b), \quad \hat{P}_a^b = Dec(Z_a^b). \tag{4}$$

Here, two decoders share the same network parameters and both are denoted as $Dec$.

## 3.3 Weakly-supervised Salient Posture Detector

Salient postures are the postures with a large range of motion, which are closely related to strong semantics of the speech content. Therefore, we propose a weakly-supervised salient posture detector to predict the saliency score of poses for each frame, which will then be used to reweight the consistency loss. This strategy enforces a more accurate alignment of salient pose representation and corresponding audio representation in the joint embedding. The architecture of the detector is depicted in Figure 3. First, we utilize the inflated ConvNet [7] to extract the initial frame-level feature $X \in \mathbb{R}^{T \times D_1}$, where $D_1$ denotes the dimension of the initial feature. Then, we feed $X$ into a temporal relation module to capture long-range temporal dependencies among features and transform $X$ into the interaction feature $Y \in \mathbb{R}^{T \times D_1}$. Then, a classifier is used to project the interaction feature into the 1D vector space to obtain the frame-level saliency score of poses $S^b \in \mathbb{R}^T$. We use the average saliency score of $top$-$k$ frames as the saliency prediction for the video sequence, and make it close to the video-sequence-level saliency label using the binary cross-entropy loss.

*3.3.1 Sequence-level Salient Label.* To facilitate the network to adaptively learn which frames are salient, we train the detector under the weak supervision of the video-sequence-level saliency labels. To acquire the sequence-level labels, we first derive the resting pose (the most frequent posture) of each speaker in the entire dataset as illustrated in the S2G [13]. For a pose sequence, we calculate the $L_2$ distance between the pose of each frame and the resting pose. Then we can obtain the probability distribution of all distances and determine whether the sequence contains salient data points based on the triple variance criterion. Video sequences containing salient points are assigned label 1 and otherwise 0.

Additionally, the ground truth of poses is pseudo and obtained using OpenPose [5], therefore it is inaccurate to directly take the saliency points as the frame-level labels. Sequence-level saliency labels can mitigate this error and facilitate the detector and generation network to learn the exact correspondence between salient postures and speech semantics. See the comparison results in Table 6 of the ablation study.

*3.3.2 Temporal Relation Module.* Inspired by the temporal attention architecture [34], we design the temporal relation module (TRM) to mine the most relevant information between frames to facilitate salient posture detection. Compared with [34], our TRM leverages the global information of a sequence instead of local information in neighbor frames to exploit long-range temporal dependencies, and capture the index prior of all frames to improve the representation capability.

Specifically, we first compute the adjacency matrix for the initial frame-level feature $X$ to measure the affinity in the embedding space, and then normalize the matrix using softmax function along each row. We denote the normalized adjacency matrix as weight matrix $W_1$, which can be formulated as:

$$W_1 = softmax(\theta(X)^\top \varphi(X)), \tag{5}$$

where $\theta$ and $\varphi$ denote linear transformation functions.

Then we incorporate the frame index prior into the network to distinguish the importance of neighbor frames to the current frame. We denote the index prior matrix as $I = [i_1, i_2, \ldots, i_t, \ldots, i_T]$, where $i_t$ is the negative relative distance vector between the current frame ($t^{\text{th}}$) and all frames. Since frames closer to the current frame have more relevant information, we use the softmax function to transform the index matrix $I$ into another weight matrix $W_2$, where closer frames are mapped to higher weights. We then concatenate $W_2$ with $W_1$ to compute the final interaction feature $Y$:

$$Y = \phi((W_1 \oplus W_2) \times X), \tag{6}$$

where $\phi$ denotes the transformation functions implemented by FC layers.

*3.3.3 Emphasizing Consistency for Salient Posture.* After we obtain the frame-level saliency score for body poses, we use the saliency score to reweight the consistency loss in Eq.(3) to enforce stronger semantic consistency for salient postures (as shown in the

upper part of Figure 2). The reweighted consistency loss can be re-formulated as follows:

$$L_{con} = \sum_{t=1}^{T} S_t^b \cdot (1 - \frac{Z_{p,t}^b \cdot Z_{a,t}^b}{\max(||Z_{p,t}^b|| \cdot ||Z_{a,t}^b||, \epsilon)}), \quad (7)$$

where $S_t^b$ is the predicted saliency score for the body pose in the $t$-th frame. The reweighted consistency loss with saliency score can effectively enforce the model to focus more on learning the mapping between salient pose and the high-level semantics of audio during the joint training.

## 3.4 Separate Face and Body Synthesis

*3.4.1 Separate Audio Feature for Face and Body.* Synthesizing facial expressions and body gestures as a whole often leads to poor lip synchronization, since different audio features are required for the generation of face and body parts: articulation-related acoustic features are required for face and semantic-related acoustic features for body. Therefore, we extract audio features dedicated to facial expressions and body gestures separately, and synthesize facial expressions and body gestures with separate branches. As shown in the lower part of Figure 2, for the face synthesis branch, we utilize a separate audio encoder and UNet to extract the rhythmic representation of audio $Z_a^f \in \mathbb{R}^{T \times D}$. Then, we decode $Z_a^f$ to generate a sequence of $T$ frames of facial keypoints $\hat{P}_a^f$ using a decoder, which has the same network structure as the body synthesis branch but different parameters.

*3.4.2 Face-Body Feature Alignment.* Although the separate body and face branches can yield individual prediction results well aligned with audio, the final gesture sequences generated by direct concatenation of the two results tend to be inconsistent and unnatural. Therefore, we design a binary classification task at the feature level to enforce the temporal alignment between representations of the body branch and face branch. As shown in Figure 4, from the semantic-related acoustic feature $Z_a^b$ (for body gesture synthesis) and the articulation-related acoustic feature $Z_a^f$ (for facial expression synthesis), we randomly sample body feature sequence $Z_{a,t_b}^b$ and face feature sequence $Z_{a,t_f}^f$, which have the same length of frames $T_c$. Here, $t_b$ and $t_f$ represent the starting index of the feature sequences respectively. We denote the one-hot label as $c \in \mathbb{R}^2$ to distinguish whether the audio features corresponding to the body and face parts are well aligned over the temporal dimension. Then, we can obtain the labeled feature pairs including the aligned positive pairs ( if $t_b = t_f$, then $c = [1, 0]$ ) and the unaligned negative pairs ( if $t_b \neq t_f$, then $c = [0, 1]$ ). We take the labeled feature pairs as training data to optimize the binary classifier $C$ to determine whether the two input features are temporally aligned. As depicted in Figure 4, the structure of $C$ consists of a GRU and three FC layers. With this module, we can align the feature space of both branches by self-supervised learning, resulting in realistic gesture generation with better synchrony. Formally, the loss function of the classification task can be formulated as:

$$L_c = \sum_{t_b, t_f \in [1, T-T_c]} -c \log C(Z_{a,t_b}^b, Z_{a,t_f}^f), \quad (8)$$

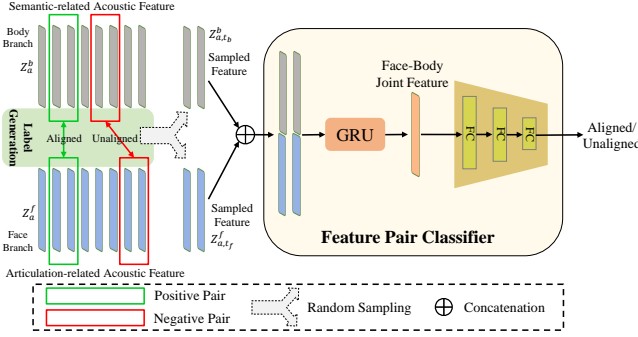

**Figure 4: The detailed structure of the face-body feature alignment module, which is trained by a self-supervised manner.**

where $C(*, *)$ is the classification result of classifier $C$.

## 3.5 Objective Function

**Reconstruction Loss.** The goal of our reconstruction path of the body synthesis branch is to learn the pose representation space by recovering the input body pose sequence. Here, we use $L_1$ loss function to measure the distance between the reconstructed body poses $\hat{P}_P^b$ and ground truth pose sequences $P^b$:

$$L_{recon} = \frac{1}{T} \sum_{t=1}^{T} ||\hat{P}_{p,t}^b - P_t^b||_1. \quad (9)$$

**Regression Loss.** The main supervision of the training process is imposed on the pose keypoints, including the supervision of the overall keypoints as well as the individual part keypoints. The supervision on the overall pose keypoints is implemented as a $L_1$ regression loss between the entire gestures $\hat{P}_a$ generated from the given audio and ground truth gestures $P$:

$$L_{reg} = \frac{1}{T} \sum_{t=1}^{T} ||\hat{P}_{a,t} - P_t||_1. \quad (10)$$

Besides, we additionally apply separate constraints on face pose keypoints and body pose keypoints by measuring the distance between gestures $\hat{P}_a^b, \hat{P}_a^f$ generated from the given audio, and ground truth $P^b, P^f$ with Huber loss ($HL$) [17]:

$$L_{body} = \sum_{t=1}^{T} HL(\hat{P}_{a,t}^b, P_t^b), \quad L_{face} = \sum_{t=1}^{T} HL(\hat{P}_{a,t}^f, P_t^f). \quad (11)$$

Then the entire loss of face part and body part is calculated as the mean value of $L_{face}$ and $L_{body}$:

$$L_{huber} = \frac{1}{2T}(L_{body} + L_{face}). \quad (12)$$

Overall, the total training objective function is:

$$\begin{aligned} L_{total} = &\lambda_r L_{recon} + \lambda_{reg} L_{reg} + \lambda_h L_{huber} \\ &+ \lambda_{con} L_{con} + \lambda_c L_c, \end{aligned} \quad (13)$$

where $\lambda_r, \lambda_{reg}, \lambda_h, \lambda_{con}$, and $\lambda_c$ are hyper-parameters that can be adjusted to control the relative significance of each loss term.

**Table 1: Quantitative comparison of all methods on four speakers (Oliver and Kubinec are from the S2G [13] dataset, Xing and Luo are collected by SDT [29]). We use $L_2$ dist. and FGD metrics to evaluate the accuracy and realism of generated results. BC and PSD metrics are utilized to evaluate the synchronization.**

| Methods | Oliver | | | | Kubinec | | | | Xing | | | | Luo | | | |
|---|---|---|---|---|---|---|---|---|---|---|---|---|---|---|---|---|
| | $L_2$ dist. ↓ | FGD↓ | BC↑ | PSD↓ | $L_2$ dist. ↓ | FGD↓ | BC↑ | PSD↓ | $L_2$ dist. ↓ | FGD↓ | BC↑ | PSD↓ | $L_2$ dist. ↓ | FGD↓ | BC↑ | PSD↓ |
| Audio2Body [32] | 49.7 | 3.48 | 0.27 | 6.29 | 70.9 | 4.51 | 0.23 | 6.20 | 50.9 | 4.75 | 0.19 | 6.41 | **48.4** | 2.70 | 0.21 | 7.81 |
| S2G [13] | 53.5 | 8.30 | 0.63 | 5.97 | 64.9 | 4.53 | 0.61 | 6.33 | 48.0 | 4.49 | 0.65 | 6.25 | 63.7 | 3.10 | 0.57 | 8.12 |
| MoGlow [3] | 50.6 | 2.28 | 0.32 | 6.23 | 78.1 | 2.49 | 0.29 | 6.39 | 48.4 | 4.94 | 0.35 | 6.34 | 54.8 | 1.47 | 0.30 | 7.86 |
| SDT [29] | 62.4 | 0.92 | 0.63 | 6.12 | 100.7 | 1.07 | 0.65 | 6.37 | 57.8 | 1.72 | 0.68 | 6.52 | 80.8 | 0.69 | 0.59 | 7.97 |
| SEEG [21] | 52.9 | 0.83 | 0.55 | 6.58 | 63.6 | 1.60 | 0.53 | 5.98 | 47.1 | 1.89 | 0.56 | 6.40 | 62.4 | 0.82 | 0.49 | 7.24 |
| DiffGesture [38] | 48.2 | 0.76 | 0.71 | 6.01 | 57.0 | 1.33 | 0.68 | 5.72 | 40.6 | 1.95 | 0.73 | 5.98 | 50.0 | 0.75 | 0.64 | 7.33 |
| Ours | **35.7** | **0.55** | **0.78** | 5.83 | **42.6** | **0.46** | 0.72 | 5.69 | 37.9 | 1.56 | 0.76 | 5.91 | 51.4 | **0.49** | 0.68 | 6.95 |

**Table 2: Quantitative results on TED Expressive dataset. We compare the proposed method with other recent methods under FGD, BC, and PSD metrics.**

| Methods | TED Expressive | | |
|---|---|---|---|
| | FGD↓ | BC↑ | PSD ↓ |
| Attention Seq2Seq [36] | 54.92 | 0.15 | 7.58 |
| S2G [13] | 54.65 | 0.68 | 7.95 |
| Joint Embedding [2] | 64.56 | 0.13 | 7.52 |
| Trimodal [35] | 12.61 | 0.56 | 7.89 |
| HA2G [24] | 5.31 | 0.64 | 7.11 |
| DiffGesture [38] | 2.60 | **0.72** | 6.97 |
| Ours | **2.50** | 0.68 | **6.83** |

## 4 EXPERIMENTS

### 4.1 Datasets

**Speech2Gesture.** Speech2Gesture [13] is a speaker-specific dataset with full body and face keypoints annotations. Following SDT [29], we test our method on four speakers: Oliver, Kubinec, Luo, and Xing. The number of different videos of four speakers is 113, 274, 72, and 27, with a total length of about 25 hours. All pseudo keypoint annotations are obtained by OpenPose [5], which contains 121 keypoints. We divide them into different parts, including 70 keypoints for the facial expression, 51 keypoints for the body pose.
**TED Expressive.** TED Expressive dataset [24] is derived from a large-scale 3D gesture dataset TED Gesture [35]. Compared to TED Gesture only with 10 upper body key points, TED Expressive is more realistic and expressive of both body and finger movements. Through 3D pose estimator ExPose [9] extracting the pose information, TED Expressive contains the 3D coordinate annotations of 43 keypoints, including 13 upper body joints and 30 finger joints.

### 4.2 Implementation Details

We use the data pre-processing protocol in SDT [29] to partition videos of 15 FPS into segments with 64 frames for training. We use a 1-layer GRU as our pose feature extractor with the hidden size of 1024 and the dimension $D$ of our audio-pose joint embedding space is 512. For the salient posture detector module, the dimension $D_1$ and $D_2$ of initial feature and interaction feature are set to 512 and 1024. In addition, we set the value of $top\text{-}k$ to 16. For both training and testing, we use a batch size of 32. We train our model with an Adam optimizer and the learning rate is set to 0.0001. For the hyperparameters, we empirically set $\lambda_r = 10$, $\lambda_{reg} = 10$, $\lambda_h = 20$, $\lambda_{con} = 1$, and $\lambda_c = 1$, which work well for both datasets.

## 4.3 Quantitative Evaluation

*4.3.1 Evaluation Metrics.* $L_2$ **Distance** is commonly used to measure the distance between the generated gestures and ground truth. **Fréchet Gesture Distance (FGD)** is proposed by [35] to measure the distribution distance of ground truth gestures and generated ones in the latent space. Note that we use the same pose feature extractor as [29] for a fair comparison.
**Beat Consistency Score (BC)** is originally proposed by [24, 38] to measure the beat correlation between audio and gesture. They utilize the angle changes of bones to quantify the motion beat and calculate the average distance between audio beat and its nearest motion beat as Beat Consistency Score.
**Pose-Sync Distance (PSD)** is designed by us to evaluate the consistency of audio and generated pose sequence. Inspired by Sync-Net [10], we train a Pose-SyncNet using a similar contrastive loss between the representations of audio and generated pose. We compute this evaluation metric using the $L_2$ distance between the audio embedding and pose embedding of our pre-trained Pose-SyncNet. Concretely, for a pose sequence $\mathbf{p}$ of 64 frames with corresponding audio $\mathbf{a}$, we evenly divide the pose sequence into $N$ pose clips $p_i$ of 9 frames and audio clips $a_i$ of 0.6$s$, and encode them using Pose-SyncNet to obtain corresponding pose feature $f_{p_i}$ and audio feature $f_{a_i}$. Then, we calculate the PSD metric as follows:

$$PSD = \frac{1}{N} \sum_{(p_i, a_i) \in (\mathbf{p}, \mathbf{a})} ||f_{p_i} - f_{a_i}||_2. \tag{14}$$

*4.3.2 Evaluation Results.* We make a comprehensive performance comparison between our method and other state-of-the-art methods on both datasets using the above metrics. As shown in Table 1 and Table 2, our method almost achieves the highest performance in all evaluation metrics, which demonstrates great superiority over existing methods. In terms of $L_2$ distance and FGD metrics, our method outperforms Audio2Body [32], S2G [13], and MoGlow [3] by a large margin, indicating that our method can produce more accurate prediction results while better maintaining the realism and diversity of the generated gestures. The incorporation of joint-embedding space and salient gesture detection can effectively enforce the semantic consistency between audio and gesture, which is critical for generating vivid and realistic co-speech gestures. Compared with SDT [29], SEEG [21], and HA2G [24], our method can generate gestures with better synchronization for both the lip motions and body pose movements, resulting in the higher BC and lower PSD. This is due to the separate audio feature extraction and synthesis

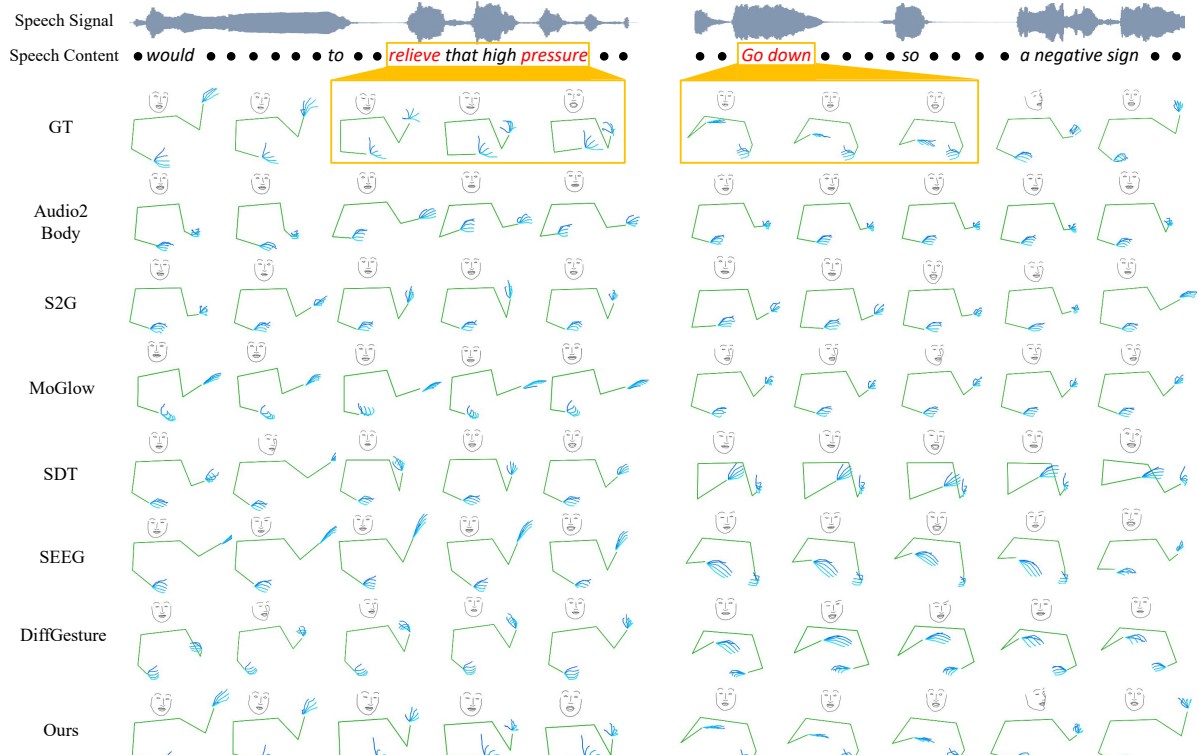

**Figure 5: Visualization results of generated gesture sequence of all methods given the speech signal. Our method can synthesize more natural and realistic gestures with better synchrony than others.**

**Table 3: User study results of the gesture sequences generated by different methods on naturalness, synchronization, and expressiveness. The rating scores range from highest (7) to lowest (1).**

| Methods | Expressiveness | Naturalness | Synchronization |
|---|---|---|---|
| GT | 6.56 | 6.62 | 6.37 |
| Audio2Body [32] | 1.18 | 2.88 | 1.34 |
| S2G [13] | 2.74 | 1.65 | 3.82 |
| MoGlow [3] | 4.16 | 3.90 | 2.26 |
| SDT [29] | 3.45 | 3.25 | 3.05 |
| SEEG [21] | 4.02 | 4.72 | 4.30 |
| DiffGesture [38] | 5.15 | 5.36 | 5.04 |
| Ours | 5.89 | 6.01 | 6.12 |

branch for facial expressions and body gestures. Overall speaking, with the combination of the joint manifold training and separate synthesizing, our method generates the most plausible co-speech gestures and achieves a better balance between synchronization and versatility.

## 4.4 Qualitative Evaluation

*4.4.1 Visualization Analysis.* As shown in Fig. 5, we visualize the generated gestures of all methods given the speech signal and compare our method with other methods. In the left case of Fig. 5, when the speaker says the verb phrase *relieve pressure*, he folds his

open arms down to the lower right, making a gathering movement to express the meaning of the phrase. Only our method successfully synthesizes this strong semantic-related gesture since we emphasize the semantic consistency of salient posture in our method. The naturalness of gestures generated by Audio2Body [32], S2G [13] and SEEG [21] is not good enough, especially the realism of hand motion generated by S2G [13] is visually poor and the face shape generated by SEEG [21] is distorted and deformed. MoGlow [15], SDT [29], and DiffGesture [38] generate less diversity in the poses. In contrast, the distribution of poses generated by our method is much closer to the real distribution. In addition, for the right case of Fig. 5, the speaker lowers his raised right hand when he says the phrase *go down*. Audio2Body [32] and MoGlow [15] generate gestures of similar appearance with a small range of motion and unnatural hands. The template of SDT [29] restricts it from generating poses with large variation, thus failing to learn strong semantic gestures. SEEG [21] and DiffGesture [38] generate the coarse pose appearance, but the generated right hand is sagging when the one of ground truth is flat. Compared with these methods, our approach can learn the salient posture and generate more realistic results.

*4.4.2 User Study.* We conduct a subjective user study to compare our method with other baselines from three aspects: naturalness, synchronization, and expressiveness. Here, naturalness refers to the smoothness and realism of the generated pose movements. Synchronization means the temporal consistency between lip motion and audio. Expressiveness measures the capability of gestures to

**Table 4: Effectiveness of key components of our framework. *Separate, Joint,* and *Detector* respectively denote separate synthesis, joint manifold space, and salient posture detector.**

| Baseline | *Separate* | *Joint* | *Detector* | FGD ↓ | BC ↑ | PSD ↓ |
|---|---|---|---|---|---|---|
| ✓ |  |  |  | 4.27 | 0.53 | 6.56 |
| ✓ | ✓ |  |  | 3.95 | 0.68 | 6.33 |
| ✓ | ✓ | ✓ |  | 1.78 | 0.71 | 6.02 |
| ✓ | ✓ | ✓ | ✓ | **0.46** | **0.72** | **5.69** |

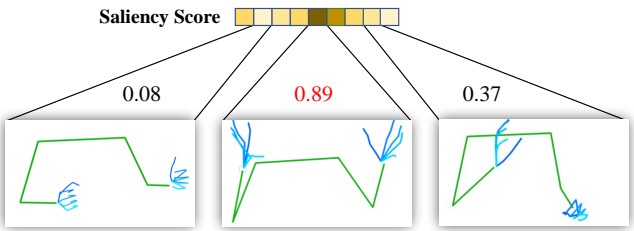

**Figure 6: The illustration of the predicted saliency score of postures. In the saliency score vector, the higher the score, the more salient the current posture.**

express the semantic information of speech. We collect anonymous gesture videos of all methods for four characters in the dataset and invite 25 volunteers to watch and score them based on the three metrics described above. The rating scores range from 1 to 7, with 7 being the most plausible and 1 being the least plausible. As shown in Table 3, our method shows significant advantages over other approaches and even achieves comparable results against ground truth. Compared with Audio2Body [32] and S2G [13], our method performs better on the gesture naturalness and synchronization due to the separate face-body synthesis framework. In addition, the integration of salient posture detection enhances the expressiveness of our method, which outperforms MoGlow [3], SDT [29], SEEG [21], and DiffGesture [38] by a large margin.

## 4.5 Ablation Study

We conduct extensive ablation studies to justify the contribution of key components to the final performance of the proposed method.
**Effectiveness of key components.** Table 4 summarizes the performance and effectiveness of different design components on the speaker of Kubinec. We use the baseline model for a fair comparison with other variants of our method, which only contains an audio encoder and pose decoder and directly predicts the holistic gesture sequence using the audio representation. From the results of the first and second row in Table 4, we can see that compared with the baseline, the integration of separate synthesis for facial expressions can facilitate the significant improvement of synchronization (See the obvious increase of BC metric). In addition, our model learns a joint manifold space to exploit the inherent semantic association between speech content and gesture, which helps to achieve the lower FGD metric and effectively enhance the realism of generated gestures. As shown in the last two rows in Table 4, the incorporation of the salient detector module can further decrease the FGD metric with +1.32 and also achieves an obvious performance boost

**Table 5: Ablation of face-body feature alignment design. W/o alignment indicates our full method without the assistance of face-body feature alignment.**

| Methods | Oliver | | | Kubinec | | |
|---|---|---|---|---|---|---|
|  | FGD ↓ | BC ↑ | PSD ↓ | FGD ↓ | BC ↑ | PSD ↓ |
| Ours | **0.55** | **0.78** | **5.83** | **0.46** | **0.72** | **5.69** |
| -w/o alignment | 0.73 | 0.70 | 5.97 | 0.89 | 0.67 | 5.81 |

**Table 6: Impact of different salient label strategies.**

| Strategy | Oliver | | | Kubinec | | |
|---|---|---|---|---|---|---|
|  | FGD ↓ | BC ↑ | PSD ↓ | FGD ↓ | BC ↑ | PSD ↓ |
| Frame-level | 1.94 | 0.73 | 6.30 | 2.01 | 0.70 | 6.25 |
| Squence-level | **0.55** | **0.78** | **5.83** | **0.46** | **0.72** | **5.69** |

in the PSD metric. Additionally, we visualize the predicted saliency score of an input gesture sequence in Figure 6.
**Ablation of face-body feature alignment.** Here, we also investigate the contribution of the face-body feature alignment design to our overall framework. From the result comparison of Table 5 on the speakers of Oliver and Kubinec, we can see that the integration of face-body feature alignment can facilitate the obvious improvement of audio-motion synchronization (see the increase of the BC metric). Meanwhile, better synchrony further helps to achieve the lower FGD metric, which indicates that the alignment module also contributes to the more realistic generated results.
**Impact of different salient label strategies.** We report the performance of different salient label strategies using the same baseline on Oliver and Kubinec in Table 6. Frame-level strategy means training the salient posture detector under the supervision of frame-level salient labels. We acquire the frame-level labels according to the distances between the poses of all frames in a sequence and the resting pose. As shown in Table 6, we observe that compared to the frame-level strategy, the proposed sequence-level approach can boost performance, especially in terms of FGD and PSD metrics. This is because sequence-level weak supervision facilitates the detector and generation network to learn the exact correspondence between salient postures and speech content, which contributes to the enhancement of realism of generated gestures.

## 5 CONCLUSION

In this paper, we propose a novel co-speech gesture generation method to enhance the learning of cross-modal association of speech and gesture. Our model learns a joint manifold space for different representations of audio and body pose to exploit the inherent association between two modalities and enforce semantic consistency using a consistency loss. Further, we introduce a weakly-supervised salient posture detector to facilitate the model to focus more on learning the mapping of salient postures and corresponding audios with highly semantic information. Extensive experiments demonstrate that the proposed method surpasses state-of-the-arts by a large margin and can effectively enhance the naturalness and fidelity of generated gestures.

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
