# OpenReview forum: "Emphasizing Semantic Consistency of Salient Posture for Speech-Driven Gesture Generation"
_acmmm.org/ACMMM/2024/Conference — MM2024 Poster_

### Official Review · Reviewer_r4aP · 2024-05-24

**Rating:** 2
**Confidence:** 4

**Summary:**

The paper presents a method to generate co-speech 3D face and upper-body gestures by explicitly focusing on salient postures and generating them accurately. The authors define salient postures as those postures consisting of large movements and are assumed to be correlated with speech semantics. They propose a consistency loss to better learn the salient postures. The consistency loss relies on weak supervision of saliency provided at the clip level – a clip has a saliency value of 1 if any of its constituent frames is determined to be salient (having large movements) and 0 otherwise. The authors also separately extract articulation-related and semantic-related features from the audio to respectively guide face and body generation. They evaluate the quantitative and qualitative performance of their proposed method through various evaluation metrics, ablation studies, visualizations, and user studies.

**Strengths:**

1. The end-to-end approach, particularly the proposed temporal relation module, is clearly explained. The corresponding ablation studies show the achieved benefits.

2. The reported experiments show good performance of the proposed method. The visual results have reasonable quality in terms of plausibility and expressiveness.

**Limitations:**

1. Missing related work: the authors have not mentioned related methods for gesture generation, such as [A, B], which generate both face and body gestures with various speaker-aware style information that may subsume the notion of posture saliency, [C] which generates emotionally expressive gestures, [D], which encapsulates the notion of posture saliency through speech rhythms, and [E, F], which achieve a wide diversity in generated gestures by using a diffusion-based approach. Given the substantial body of work that the authors have not mentioned, it is hard to fully understand their proposed benefits.

[A] Habibie, Ikhsanul, Weipeng Xu, Dushyant Mehta, Lingjie Liu, Hans-Peter Seidel, Gerard Pons-Moll, Mohamed Elgharib, and Christian Theobalt. "Learning speech-driven 3d conversational gestures from video." In Proceedings of the 21st ACM International Conference on Intelligent Virtual Agents, pp. 101-108. 2021.

[B] Habibie, Ikhsanul, Mohamed Elgharib, Kripasindhu Sarkar, Ahsan Abdullah, Simbarashe Nyatsanga, Michael Neff, and Christian Theobalt. "A motion matching-based framework for controllable gesture synthesis from speech." In ACM SIGGRAPH 2022 Conference Proceedings, pp. 1-9. 2022.

[C] Bhattacharya, Uttaran, Elizabeth Childs, Nicholas Rewkowski, and Dinesh Manocha. "Speech2affectivegestures: Synthesizing co-speech gestures with generative adversarial affective expression learning." In Proceedings of the 29th ACM International Conference on Multimedia, pp. 2027-2036. 2021.

[D] Ao, Tenglong, Qingzhe Gao, Yuke Lou, Baoquan Chen, and Libin Liu. "Rhythmic gesticulator: Rhythm-aware co-speech gesture synthesis with hierarchical neural embeddings." ACM Transactions on Graphics (TOG) 41, no. 6 (2022): 1-19.

[E] Ao, Tenglong, Zeyi Zhang, and Libin Liu. "Gesturediffuclip: Gesture diffusion model with clip latents." ACM Transactions on Graphics (TOG) 42, no. 4 (2023): 1-18.

[F] Yang, Sicheng, Zhiyong Wu, Minglei Li, Zhensong Zhang, Lei Hao, Weihong Bao, Ming Cheng, and Long Xiao. "DiffuseStyleGesture: Stylized Audio-Driven Co-Speech Gesture Generation with Diffusion Models."

2. Further, there are more recent datasets for co-speech face and body generation, such as [G], which is commonly used by recent approaches. Without any results on more recent datasets, it becomes harder to understand the proposed benefits over available methods.

[G] Ghorbani, Saeed, Ylva Ferstl, Daniel Holden, Nikolaus F. Troje, and Marc‐André Carbonneau. "ZeroEGGS: Zero‐shot Example‐based Gesture Generation from Speech." In Computer Graphics Forum, vol. 42, no. 1, pp. 206-216. 2023. https://github.com/ubisoft/ubisoft-laforge-ZeroEGGS

3. Do the authors have any evidence, such as experiments or prior studies, to support their claim that faces are more correlated with articulation-based audio features and poses are more correlated with semantic-related audio features? This is a key aspect of their network design, but it is not fully justified.

4. Have the authors considered any intermediate approach between assigning the saliency labels at the frame level and at the clip level? For example, assigning the saliency value for a contiguous group of frames where there are significant movements? The approach of assigning entire clips as salient appears too loose a constraint.

5. Why does the label to determine alignment between face and body gestures (Line 506) consist of two elements? It seems only the values $[1, 0]$ and $[0, 1]$ are used, so why not use only $0$ and $1$?

**Suitability:**

3

---

### Official Review · Reviewer_92MD · 2024-05-24

**Rating:** 4
**Confidence:** 4

**Summary:**

This paper aims to generate upper body gestures and facial expressions. The main target is generate more "Salient Posture" - gestures with  larger motion ranges. The authors introduce a pipeline in four steps: 1) learn a audio-gesture joint embedding by minimizing the cosine similiarity between audio and motion latent space. 2) label the pose with larger L2 distance with mean pose as Salient Posture, and train a classifer to learn this label. 3) use this learned classifer to generate pseudo label during training, give the predicted Salient Posture a higher weight 4) use a face and body align module for final results. Results conduct on TED and S2G shows a large improvement.

**Strengths:**

1. the idea of using a weakly-supervised method to geneate semantic aware pose is more easier for scaling up for real world application.
2. experiments compare enough methods, objective results are good.

**Limitations:**

I feel postive for this paper, however, I have a lot of unclear points for the motivation and correctness of the pipeline design.

1. motivation of joint embedding for generation.
previous method directly learn audio2gesture, the authors add gesture reconstruction and align the latents. the reconstrcution loss and latent cosine similiarity loss is trained jointly. why this would be better? Previous methods like TalkShow, first learns the reconstruction of gesture is in a pretrained VQVAE  / AE, then map the audio to the motions latent, this speerated training seems better.

2. what is acturally we are doing with L2-distance based Salient Posture Classifier?
Salient Posture Classifier are trained with labels based on the L2-distance to mean pose (or it is like a mean pose),  so acturally we are giving the motion with higer L2-distance a higher loss weight, if so, why we do not directly resample the training data based on their L2-distance like DisCo? or why we do not directly calculate generate results - mean pose as a loss weight? The weakly supervised idea is not so bad, but do we need it?

3. the resolution of cosine similarity calculation,
From the paper, it seems the cosine similiarity is calculated per-frame between audio and gesture latents. Is this correct? I could understand the frame-level similiarity will focus on aligning the rhythmic features but cannot get how it align semantic. As the semantic similarity may activated in particular frames, not all frames.

Minor issue:
A lot of related works are missing in the context of co-speech gesture generation such as previous MM papers. Only one new method after 2021-DiffGesture-is discussed in this paper.

**Suitability:**

3

---

### Official Review · Reviewer_Y2in · 2024-05-26

**Rating:** 4
**Confidence:** 3

**Summary:**

The existing co-speech gestures generation only considers the mapping from audio to the body gestures. In this paper, the authors emphasize the semantic consistency between the speech content and postures. To achieve this, they proposed that the salient posture is highly related to some specific phrase. Therefore, they enforce semantic consistency by emphasizing the salient postures that are detected by a weekly-supervised detector. In addition, the authors treat the body language and facial expressions independently compared to the previous work. The qualitative results show the model can generate more diverse and semantic-related co-speech gestures.

**Strengths:**

1. The authors take the speech content into consideration and enforce the semantic consistency between the speech and generated gestures. Although the direct alignment between posture and speech content is challenging, they improve the generation by emphasizing the salient postures.
2. The attached video demonstrates that the independently generated facial expression is of a higher quality compared with the baselines.

**Limitations:**

1. The example of phase "cooling down" shows the correlation between speech content and salient postures. However, it is hard to say that the slight postures are trivial.
2. Although the salient posture is emphasized during training, how do you guarantee the generated salient gesture is semantically correct? For example, a rising movement might be generated with the phrase "cool down"

**Suitability:**

3

---

### Meta-Review · Area_Chair_t9tB · 2024-07-06

**Recommendation:** Accept (Poster)
**Confidence:** 5

**Metareview:**

All the reviewers have acknowledged the contributions of this paper. The paper is technically sound and a great fit for MM. The authors did a good job addressing the points raised by the reviewers. However, as noted by Reviewer r4aP, there are still unresolved issues. The authors should elaborate on the related work and discuss their contributions with respect to the existing recent approaches adequately, as they promised to do so in their rebuttal.